# Reproductive Anatomy of Chondrichthyans: Notes on Specimen Handling and Sperm Extraction. I. Rays and Skates

**DOI:** 10.3390/ani11071888

**Published:** 2021-06-25

**Authors:** Pablo García-Salinas, Victor Gallego, Juan F. Asturiano

**Affiliations:** 1Grupo de Acuicultura y Biodiversidad, Instituto de Ciencia y Tecnología Animal, Universitat Politècnica de València, 46022 Valencia, Spain; pablo.g.salinas@outlook.com (P.G.-S.); vicgalal@upvnet.upv.es (V.G.); 2Associació LAMNA per a l’estudi dels elasmobranquis a la Comunitat Valenciana, Fraules 10, 46020 Valencia, Spain

**Keywords:** fish reproduction, aquarium, ex situ conservation, reproductive assisted techniques, Batoidea, artificial insemination

## Abstract

**Simple Summary:**

Many species of rays and skates are endangered, and ex situ conservation programs developed by research centers and public aquaria could improve this situation. To reproduce these species in captivity, scientists need to know how to extract their sperm and how to conduct their artificial insemination; however, the anatomical diversity of the reproductive organs of this group of animals is a handicap. A comparison of the reproductive anatomies of 11 distinct species is presented here, emphasizing the important differences between the species when performing sperm extraction or artificial insemination. In addition, the process of obtaining sperm samples from both live and dead males is described in detail, using both cannulation and abdominal massage.

**Abstract:**

The superorder Batoidea (rays, skates, and relatives), constitutes one of the most threatened group of vertebrates. Strengthening ex situ conservation programs developed in research centers and public aquaria could be a way of addressing this situation. However, captive breeding programs must be improved to prevent the capture of wild animals and to develop proper in situ reintroduction strategies. Sperm extraction and artificial insemination are two techniques commonly used in other threatened species, which could also be used in rays and the like. However, the different reproductive morphologies present within this group of animals may hamper both processes. Here, we present a comparison of the reproductive anatomies of 11 distinct batoid species, emphasizing the important differences between the species when performing sperm extraction or artificial insemination. Both male and female animals, belonging to the Rajidae, Dasyatidae, Torpedinidae and Myliobatidae families, from the Mediterranean Sea were studied. In addition, we describe the procedure to extract sperm using both cannulation and abdominal massage, either from live or dead batoids Finally, the obtention of motile sperm recovered from the oviducal gland of females is described. These techniques generate a new range of possibilities for the conservation of these threatened species.

## 1. Introduction

Appearing almost 400 million years ago, Chondrichthyan fishes are an old and ecologically diverse group with a key role in the regulation of the ecosystems they inhabit [1,2]. The class is comprised of 1472 species [3], classically divided into the Holocephalans, commonly named chimaeras, and the Elasmobranchs, which include sharks and rays. This last group, rays and their relatives (skates, guitarfishes, sawfishes and the like), is the most diverse group among Chondrichthyan fishes, with 816 species accepted under the formal name of Batoidea [4,5,6]. Like all other elasmobranchs, batoids possess life histories that make this group sensitive to elevated anthropic pressures [7,8], and in fact, overfishing and habitat destruction are the main drivers for the global decline of their populations [9,10].

In order to understand the current global situation of the batoid populations, it is essential to know their life histories and reproductive strategies. Rays and their relatives are characterized by large body sizes, late sexual maturity, long gestation periods, high maternal investment, and reduced offspring [10,11]. Batoid reproductive strategies are highly diverse and can be categorized based on the nutrition method of the embryos. Lecithotrophic methods include oviparity (as in the case of Rajiformes) and yolk sac viviparity (as in the case of Torpediniformes), where the only nourishment comes from their yolk sack, while matrotrophic methods include an additional source of nourishment at some point in the embryo development, in the form of lipid histotrophy, also known as uterine milk (as in the case of Myliobatiformes) [12,13,14]. Like all other Chondrichthyans, fertilization in batoids is internal, with males having intromittent appendages called claspers as a modification of their pelvic fins [15].

All these reproductive factors and complex life histories have discouraged captive breeding programs in aquaculture industries [12], but not in aquaria facilities, neither public nor private. The reproduction of batoids in aquaria has been reported for some species [16,17], but breeding programs in aquaria have traditionally relied on natural mating rather than the use of reproductive techniques, such as artificial insemination [18]. This technique has been receiving increasing attention, but to ensure the success of this technique, a reliable supply of sperm is required, especially in the case of endangered species [18,19,20,21]. Sperm can be obtained from dead or live animals, and although there has been success in obtaining sperm from several batoids [18,22,23,24], the procedures of extraction may vary between the different species. Cannulation and abdominal pressure are the traditional procedures used [21], but with these techniques, the location and morphology of the species-specific reproductive structures (for example seminal vesicles and urogenital papilla) need to be considered.

The main objective of this study is to provide a useful guide of the anatomy of the reproductive system of batoids, with a particular focus on sperm procurement procedures, and to propose preliminary indications in the female anatomy to be considered during artificial insemination.

## 2. Materials and Methods

### 2.1. Origins of the Specimens

Males and females belonging to 11 batoid species of the order Rajiformes (*n* = 20), Myliobatiformes (*n* = 14) and Torpediniformes (*n* = 4) were studied (Table 1). Some of the species (rough skate *Raja radula*, Mediterranean starry skate *Raja asterias*, thornback skate *Raja clavata*, spotted skate *Raja montagui*) were available in fish markets or from commercial fishing vessels’ bycatch (longnosed skate *Dipturus oxyrinchus*, spiny butterfly ray *Gymnura altavela*, common stingray *Dasyatis pastinaca*, common eagle ray *Myliobatis aquila*, bull ray *Aetomylaeus bovinus*, marbled electric ray *Torpedo marmorata*). Others were part of the zoological collection of a public aquarium (Oceanogràfic, València). In this case, the fish (Mediterranean starry skate *Raja asterias*, undulate skate *Raja undulata*) were kept separately in two 8000 L tanks with recirculating sea water (temperature: 16–18 °C; salinity: 35–37‰) and fed twice a day with herring, squid, and shrimps. Maturity of all the specimens was determined by gonad development, clasper calcification and their size according to bibliography [25].

### 2.2. Dissection Procedure

A dissection was performed on each individual to gain access to the reproductive system of the animals. Dissection focused only on the reproductive system of the specimens, adapting the dissection protocols used by other authors for sharks [26,27,28]. Anatomical terminology used in the present study was adapted from De Iuliis and Pulerà (2019) [28]. The specimens were flipped dorsally, exposing their ventral surface (Figure 1A). Using scissors, a small incision was made over the coracoid bar of the pectoral girdle (arrowhead in Figure 1A), large enough to allow the introduction of a scalpel blade. Next, with the help of forceps to keep the abdominal wall (skin, muscle, parietal peritoneum) elevated, a cut was made along the scapular process contour towards the pelvic girdle (dotted line in Figure 1A). Thus, the tissue was removed to leave the pleuroperitoneal cavity exposed. To access the reproductive system, the liver (arrowhead in Figure 1B) and the esophagus, stomach, spiral valve, and rectum (arrowhead in Figure 1C) were removed. Hepatic lobes were drawn forward and removed by cutting through their cranial attachments (hepatic ducts and falciform ligament) (dotted line Figure 1B). Special care was taken throughout this process so as not to damage the hepatic lobes or gallbladder, and to avoid the leakage of the oil and bile present in these organs. The stomach and intestine were removed by making an incision through the esophagus and a cut through the rectum, close to the rectal gland (dotted lines in Figure 1C). To remove the digestive tract, the mesogaster and caudal mesenteries were carefully cut to avoid damage to the mesorchium. In this way, easy access to the urogenital system was achieved (Figure 1D).

To clearly expose the caudal part of the urogenital system (seminal vesicle, urogenital sinus, urogenital papilla in males, or uterine sphincters and urinary papilla in females), an incision through the puboischiadic bar was made (dotted line in Figure 1E). Furthermore, in some species, the removal of the cloacal lips to access the urogenital/urinary papillae was necessary. For a clear view of the lower portion of the ductus deferens and seminal vesicle, the parietal peritoneum (arrowhead in Figure 1F) should carefully be removed.

### 2.3. Description of Reproductive Structures

A macro lens was used to take detailed photographs of each species throughout the dissection procedure of the specimens, and illustrated notes were also taken. The main objective of each dissection was to: (i) determine how to gain easy access to the urogenital papilla, (ii) observe the number and disposition of urogenital pores, (iii) observe urogenital sinus morphology, (iv) access the seminal vesicle/uterus. Plastic tubes with different gauges (0.5–2 mm in diameter) were used as probes during the dissection to confirm the access and connections from the external part of the reproductive system (urogenital/urinary pores and papilla) to the internal part (urogenital sinus, seminal vesicle, uterus). Additionally, the information obtained was combined and used to propose a general morphological scheme for an ideal batoid, both for females and males.

### 2.4. Sperm Collection

#### 2.4.1. In Vivo Sperm Extraction

Tonic immobility was induced prior to sperm extraction to minimize struggling and reducing the stress during handling [29,30,31], by placing the animals in an upside-down position with their mouth, spiracles and gill slits submerged, while gentle pressure was applied in their snouts. Having the cloaca emerged above water, gentle pressure was applied to the abdominal area over the location of the seminal vesicle, to make the sperm flow through the urogenital papilla. The sperm was immediately collected using a sterile syringe or pipette and transferred to sterile tubes after collection.

#### 2.4.2. Post-Mortem Sperm Extraction

Animals were cleaned using marine water to remove mucus and other biological remains such as blood and fishery residues (mud and remains of other organisms which may be found in animals obtained from fisheries). Three different methods were used to obtain sperm form dead animals: (i) abdominal massage on the ventral region immediately anterior to the pelvic girdle, or by pressing around the urogenital papillae in the cloacal cavity with curved pincers; (ii) accessing the internal cavity through dissection and stripping directly on the seminal vesicle (in both cases, sperm flowing from the urogenital papilla was immediately collected using a sterile syringe or a pipette); and (iii) introducing a polyurethane cat catheter (BUSTER cat catheter, 1.0 × 130 mm^2^, Kruuse. Langeskov, Denmark) or a PVC nasogastric tube (Feeding Probe L/RX CH-05 2.67 × 50 mm^2^, JMEDIS. Cádiz, Spain) through the appropriate pore on the urogenital papilla. A sterile lubricating jelly with antiseptic (Optilube ActiveTM, Optimum Medical. Leeds, UK) was used to facilitate the introduction of the tube. Tubes and catheters were continuously rotated while inside the seminal vesicle to avoid clogging.

Oviducal glands from common stingray *D. pastinaca*, rough skate *R. radula*, Mediterranean starry skate *R. asterias*, longnosed skate *D. oxyrinchus* and marbled electric ray *T. marmorata* females were obtained (Figure 2A) during dissection by cutting through the oviduct on both sides of the gland. The organ was carefully cleaned with artificial sea water to remove blood stains from the extraction procedure (Figure 2B). Care was taken to avoid sea water flowing into the gland through the oviducts. The glands were split along the longitudinal axis, exposing the lumen (Figure 2C). Using the blunt edge of a scalpel blade, the surface of the luminal epithelium (Figure 2D) was scraped to collect a pearly mucus. The mucus was diluted in an artificial seminal plasma extender described by García-Salinas et al. (2021) [32]. The pH and osmolality of the main components of the extender (in mM; 433 Urea, 376 NaCl, 120 Trimethylamine N-oxide (TMAO), 8.4 KCl, 50 Glucose, 7 CaCl_2_-2H_2_O, 3.5 NaHCO_3_, 0.08 Na_2_SO_4_, 1.4 MgSO_4_) were adjusted to 6.5 and 1000 mOsm/kg respectively, in order to match the levels of the physiological fluids.

## 3. Results and Disscusions

### 3.1. Female General Anatomy

Although there are a wide variety of anatomical differences between the species studied, this study aims to highlight the differences in the reproductive system that may be relevant to the use of specific techniques, such as sperm extraction and artificial insemination. Other possible differences not related to this topic have not been considered. Even though abdominal pores are not part of the reproductive system, its misidentification may cause errors during cannulation, so its description is offered.

Unlike males (Figure 3), batoid females do not possess intromittent structures on their pelvic fins. The cloaca is located between the pelvic fins, and in some species is enveloped by two cloacal lips on either side, covering the anus and the urinary papilla. As the only pores in the papilla are those found in the urinary system, the term urogenital papilla should be limited to males, and not used for females. Depending on the species, one or two pores can be located on the papilla. In the cranial section of the cloaca, the apertures of the uteri can be found. In some species, there are two separated orifices (one for each uterus) easily accessible from the cloaca, whereas in other species, the orifices converge in a common cervix formed by the union of the cervices (or cervixes) of each uterus. The two abdominal pores (also known as celomic pores) are located near the cloaca closed by a sphincter. These pores connect the pleuroperitoneal cavity with the exterior and may allow the removal of fluid from the inner cavity [28].

Internally, females have one or two functional uteri [12] (depending on the species) either where the embryos develop or where the eggs are kept until they are laid. The oviducts are connected to the cephalic portion of the uteri. The oviducal gland (also known as the shell gland or nidamental gland) can be found on these narrow tracts. The functions of this gland are the storage of sperm, oocyte fertilization, and the formation of the egg case [33]. Thus, in oviparous species, the shell gland is bigger than in ovoviviparous or viviparous species. In some species, the gland also supposes the terminological division of the oviduct in posterior or anterior oviduct. A sphincter-like structure, the isthmus, can seal the oviducal gland from the posterior oviduct and the uterus in some species. The anterior oviducts converge in the ostium, a funnel-like structure through which the unfertilized oocytes coming from the ovaries pass into the uterine system. Depending on the species, the ovaries can be paired; fused in a single ovary; or vestigial, with only one ovary developed.

### 3.2. Female Comparative Anatomy

Overall, the general morphological structure of the female reproductive system is well maintained across all the species studied (Figure 4). This anatomical resemblance was particularly remarkable between skates (species from the genus *Raja*) (Figure 4A) and the marbled electric ray *T. marmorata* (Figure 4E), probably because of the close evolutionary relationship between these two plesiomorphic groups [34]. Both groups have the same disposition of the basic anatomical structures, but with a clear divergence in the size of the oviducal gland. Members of the Rajiformes order (where genera *Raja* and *Dipturus* are assigned) are oviparous [13], and the oviducal gland is therefore well developed in order to produce a hard egg case [33], while members of the order Torpediniformes are ovoviviparous (also called yolk-sac viviparous) [13], and therefore, their gland is not so well developed. The gland is also not well developed in the common stingray *D. pastinaca* or the other two species of Myliobatiformes (common stingray *M. aquila*, and bull ray *A. bovinus*). Not only are they evolutionarily close, but they share the same reproductive method (lipid histotrophy) and do not produce hard egg cases (although the fertilized eggs are grouped together and encapsulated in one thin egg case at the beginning of the development [35,36]).

Other differences can be seen in the shape of the cloaca and the position of the abdominal pores, particularly in the common stingray *D. pastinaca* (Figure 4C) and common eagle ray *M. aquila* (Figure 4B), where the abdominal pores are placed on the cloacal lips instead of on the caudal part of the cloaca. As mentioned before, the pores may help with the regulation of liquid in the celomic cavity and should not be confused with uterine sphincters or other structures related to the reproductive system. Also, in this species, the urinary papilla is considerably larger than that of the other species observed and has two independent urinary pores instead of one. Access to the uteri is closed by a sphincter, which is easy to reach with the help of tweezers when the cloacal lips are removed or separated. In the longnosed skate *D. oxyrinchus* (Figure 4F), the paired cervices (or cervixes) are fused in a wide common cervix, with the urinary papilla in one extreme and the cloaca in the other. The dimensions of this cervix can be explained by the large size of the eggs of this species (up to 14 cm in length and 10 cm in width [25]). In the other viviparous species examined (genus *Raja*, Figure 4A), a smaller cervix compared with that of *D. oxyrinchus* could be found. Animals from this genus possess considerably smaller eggs than those of the genus *Dipturus*, despite belonging to the same family Rajidae [25].

It should be noted that despite the females presenting two uteri, both are not always functional. This consideration is of special interest if an artificial insemination is intended, as mentioned below. Some species, such the common stingray *D. pastinaca*, the common eagle ray *M. aquila*, and the bull ray *A. bovinus*, showed developmental differences between the right and left ovaries. This phenomenon has also been described in other members of the same families [37,38,39,40] and could result in a non-functional right uterus. Nevertheless, in this study, both uteri of both *M. aquila* females were shown to be functional, despite the ovarian size difference, as previously described for the species [36].

### 3.3. Anatomic Notes for Artificial Insemination

To date, there are only two published cases of artificial insemination in batoids: in the clearnose skate *R. eglanteria* [19], and recently in the ocellate river stingray *Potamotrygon motoro* [20]. This technique has also been considered for the spotted eagle ray *Aetobatus narinari* [38]. In short, insemination is achieved by inserting a catheter either through the cloaca to deliver sperm into the cervix or through the uterus and into the oviducal gland, although it has been proposed that intrauterine (or oviducal) insemination results in higher fertilization rates than cervix insemination. [18,19]. To be able to deposit a sperm sample in the correct oviduct, the catheter needs to be inserted through different structures (such as cloaca, uterine sphincters or cervix), which can hamper the overall process [20]. Moreover, other factors such as the functionality of the uterus mentioned above should be considered. Thus, to perform artificial insemination protocols, the female reproductive system must be known in the same way as the morphology of the male reproductive system when extracting sperm.

### 3.4. Male General Anatomy

Externally, male batoids have paired elongated claspers (also called myxopterygia) at the base of the pelvic fins, used as intromittent organs for internal fertilization (Figure 5). Claspers are in fact a rolled up prolongation of the pelvic fins, forming a tube-like structure with a ventral groove known as the hypopyle. Through this groove, sperm is transported from the male urogenital papilla to the female cloaca and then into the uterine sphincter, although alternate sperm ducts may ejaculate semen, coating the claspers which are then inserted [12]. The clasper gland is located in the proximal part of the clasper. The function of these ovoid glands is uncertain, although it seems to be related to the process of sperm propulsion through the hypopyle and the secretion of substances to protect or activate the sperm [41,42]. As in females, two abdominal pores are located near the cloaca closed by a sphincter, connecting the pleuroperitoneal cavity to the exterior [28].

The cloaca is located between the pelvic fins, usually near the origin of the claspers. In some species, the cloaca is enveloped by two cloacal lips at either side, covering the anus and the urogenital papilla. Depending on the species, between one and four pores can be located on the urogenital papilla, connecting with the urinary and reproductive systems. In members of the genus *Raja*, a special sphincter resembling two small lips closes the ducts leading to the urogenital system.

Internally, the male reproductive system is composed of paired testes embedded in the epigonal organ located near the cephalic region of the animal, and as well as two genital tracts that can also be divided into different sections: the proximal convoluted part, which is called the epididymis; the ductus deferens, also called Wolffian duct or vas deferens [12]; and the distal widened portion of the ductus deferens, called seminal vesicle or ampulla [15], where the sperm is stored. The Leydig gland is adjacent to the ductus deferens onto which it empties its contents, as well as onto the epididymis. The gland, formed by the cranial part of the mesonephros, has a secretory function by producing seminal fluid and the matrix where the spermatozeugmata or the spermatophores are formed [13,15,43]. Another gland related to reproduction, and unique to batoids, is the alkaline gland. The gland is located in the caudal part of the body cavity and empties its content into the urogenital sinus or directly into the last portion of the seminal vesicle or ductus deferens [41]. Although the main function of the gland is still being discussed, it is known that its secretions have a positive effect on the activation of the sperm, thus increasing spermatozoa motility [19,41].

### 3.5. Male Comparative Anatomy

Some important differences were observed in the urogenital papilla morphology between the species studied that can affect the cannulation and sperm extraction procedures. Firstly, the members of the genus *Raja* studied have a sphincter similar to two small muscular lips that can block the entrance of an extraction catheter (Figure 6A). Once the sphincter is passed, the seminal vesicles converge in a sinus of small size where a scarce amount of sperm can be found, and so to perform the sperm extraction, the catheter must be diverted slightly to one side or the other to access the seminal vesicles. In this genus, seminal vesicles are not pouch-shaped, but a sinuous widening of the end of the vas deferens. The structure is fragile and the catheter should not be inserted more than 2 cm, or else there is a risk of damage. The same sinuous morphology can be observed in *D. oxyrinchus*, but in this species, there is no sphincter, and a single pore at the tip of the urogenital papilla can be found.

Unlike in the previous species, the common stingray *D. pastinaca* has three pores on the tip of its urinary papilla (Figure 6B). Two sit alongside one another and are the access to the seminal vesicles, while the third pore in the center of the papilla leads to the urinary system. When using a catheter to perform sperm extraction, the tube should be introduced through the genital pores to empty each seminal vesicle’s contents, as the seminal vesicles are independent. Seminal vesicles in *D. pastinaca* are less sinuous than in *Raja*, thus the catheter can be inserted easily. Similar morphology can be observed in the reproductive system of *G. altavela*, but it must be noted that the only specimen examined was an underdeveloped male, and some changes are possible during ontogeny.

Although being closed from an evolutionary perspective [34], some variations exist in the morphology of the urogenital system of the common eagle ray *M. aquila* (Figure 6C) and the bull ray *A. bovinus* (Figure 6D). *M. aquila* possess four different pores in the urogenital papilla, with the upper two (slightly bigger) leading to the seminal vesicles. As in *D. pastinaca*, both seminal vesicles are independent, and the catheter for obtaining sperm must be inserted in a straight line, otherwise it may swerve and end up in the alkaline gland. On the other hand, *A. bovinus* has only two pores arranged longitudinally on the tip of the urogenital papilla. The pore located above is the access to the reproductive system (seminal vesicle and alkaline gland), while the other pore leads to the excretory system. Both seminal vesicles converge in a small sinus in the urogenital papilla, but for sperm extraction purposes, they can be considered to be two independent seminal vesicles. Cannulation through the genital pore should be done using a slight angle to the left or to the right in order to reach both vesicles. Although all the males from all the species possess alkaline glands (albeit in some cases small in size, as in the case of *T. marmorata*), only those of the bull ray *A. bovinus* have been depicted, because in the other species, the presence of this gland does not suppose any obstruction during sperm collection. However, in this species, the size and location of the efferent duct from the gland resulted in some failed cannulation attempts before finally reaching the seminal vesicle.

*Torpedo marmorata* seminal vesicles converge in a sinus in the urogenital papilla with a single pore (Figure 6E). The overall position is similar to that of *Raja* or *D. oxyrinchus*, but the vas deferens and seminal vesicle are less sinuous. The urogenital papilla does not have a sphincter as in the case of *Raja*, but the cannulation to obtain sperm can be complicated because of the small size of the male electric rays compared to the females.

### 3.6. Anatomic Notes for Sperm Extraction

Abdominal massage in flat batoids (Rajiformes, Myliobatiformes, Torpediniformes) for sperm extraction can sometimes be difficult because of the shape and position of the seminal vesicles. In sharks, abdominal massage can be performed by applying pressure to the pelvic region and the sides of the body; however, the seminal vesicle of flat batoids cannot be reached laterally due the presence of pectoral fins, which hampers extraction. Thus, obtaining sperm was only possible in mature males with a large quantity of sperm in their seminal vesicles by pressing on the area around the urogenital papilla. It should be noted that when pressure is applied to the abdominal region, some of the contents of the digestive system may come out, spoiling the sample. Furthermore, the flow of sperm from the urogenital papilla to the cloaca can easily expose the samples to microbial contamination, compromising its preservation if a medium-term preservation (days or weeks) of the samples is desired.

On the other hand, cannulation is a better technique for obtaining sperm samples when the amount of sperm in the seminal vesicle is scarce, or a clean sample is required. The use of a lubricating jelly with antiseptic can improve the insertion of the cannulation tubes, reducing damage to the surrounding tissue and reducing microbial contamination. Most batoids have morphologically complex seminal vesicles and care must be taken when inserting the cannulation tubes so as not to damage the structure. Perforations in the seminal vesicle can be produced easily, resulting in the catheter accessing the retroperitoneal space and the kidney. If that is the case, blood will be easily seen inside the catheter. While performing a cannulation, the suction of the syringe must be gentle to avoid clogging the distal apertures of the cannula. Also, to prevent clogging, the cannula should be rotated constantly. The angle of insertion of the catheter should also be considered to avoid the catheterization of the alkaline gland instead of the seminal vesicle. For this reason, an angle of 0–5° in relation to the longitudinal axis is advised.

Sperm storage in the oviducal gland has been reported for several batoids, sharks and even chimaeras (for a brief review, see Marongiu et al. 2015 [44]), and is considered to be an evolutionary mechanism conserved to ensure fertilization in nomadic species or those with low population densities [45]. The storage of the sperm occurs in the gland regions called baffle and terminal zones [33,44], and Marongiu et al. (2015) state that it is possible to find sperm in other zones, perhaps as a result of recent mating events. After gentle scraping of the luminal epithelium of the oviducal glands from *D. pastinaca*, *D. oxyrinchus*, *R. radula*, *R. asterias* and *T. marmorata*, a pearly mucus was obtained and diluted in an extender solution. Aliquots from this dilution were observed under a microscope, revealing the presence of ciliated epithelial cells, cellular fragments (including spermatozoa remains) and, in four species (*D. oxyrinchus*, *R. radula*, *R. asterias* and *T. marmorata*), motile sperm were found. Although the presence of motile cells could be an indicator of recent mating and not long-term stored sperm, it should be noted that in at least one case, (*D. oxyrinchus*) developed eggs were found in the uteri, so mating could not have been very recent. Although the small concentrations of motile spermatozoa in these samples perhaps limits the use of this technique as a way to source sperm to perform artificial insemination, other studies focusing on genetics, spermatozoa morphology or motility patterns can still be developed.

Public aquaria play an important role in batoid conservation, through ex situ conservation programs [46] based on public outreach initiatives and the establishment of an emotional connection between the visitor and the animal [47], by supporting research (including reproductive methods) of different species, and by training professional staff for animal handling and sampling [21]. But more steps towards real sustainability should be made, especially when threatened species are involved [16,48]. Public aquaria should start specific breeding programs and cease relying on the spontaneous reproduction of the animals in their collections or on captures from the wild. Reproductive breeding programs using reproductive technologies, such as sperm extraction and artificial insemination, could not only aid these steps towards aquaria sustainability, but could also be the key to developing reintroduction programs back to the wild for threatened species of elasmobranchs. It should be noted that among the batoids, we can find some of the most threatened marine animals on the planet: the sawfishes, guitarfishes and wedgfishes. [49,50,51]. In fact, “responsible husbandry” is one of the objectives of the global strategy for the conservation of sawfishes [52]. Although more research on batoid reproduction must be carried out, the goal of this study was to offer a useful guide with protocols that can serve as tools for the conservation of these and other species.

## 4. Conclusions

Many threatened rays and related species could benefit from breeding programs using specific reproductive techniques. These programs could provide specimens with which to carry out conservation, either through education, research or even reintroduction into the wild. The techniques developed in this study are intended to be useful for carrying out these programs. Although abdominal massage is the simplest technique for obtaining sperm, it is not the most effective in the case of batoids, due to the peculiar morphology of their body and seminal vesicles, which are often reduced in size. Although cannulation is a more complex technique, it allows the sampler to obtain sperm in a more precise and clean way if the anatomy of the animal is known. Finally, obtaining active sperm from the oviducal gland in females opens new research opportunities that should be exploited in the future. Much work remains to be done in the development and application of reproductive techniques in batoids, but these first steps may be crucial for the future conservation of these animals.

## Figures and Tables

**Figure 1 animals-11-01888-f001:**
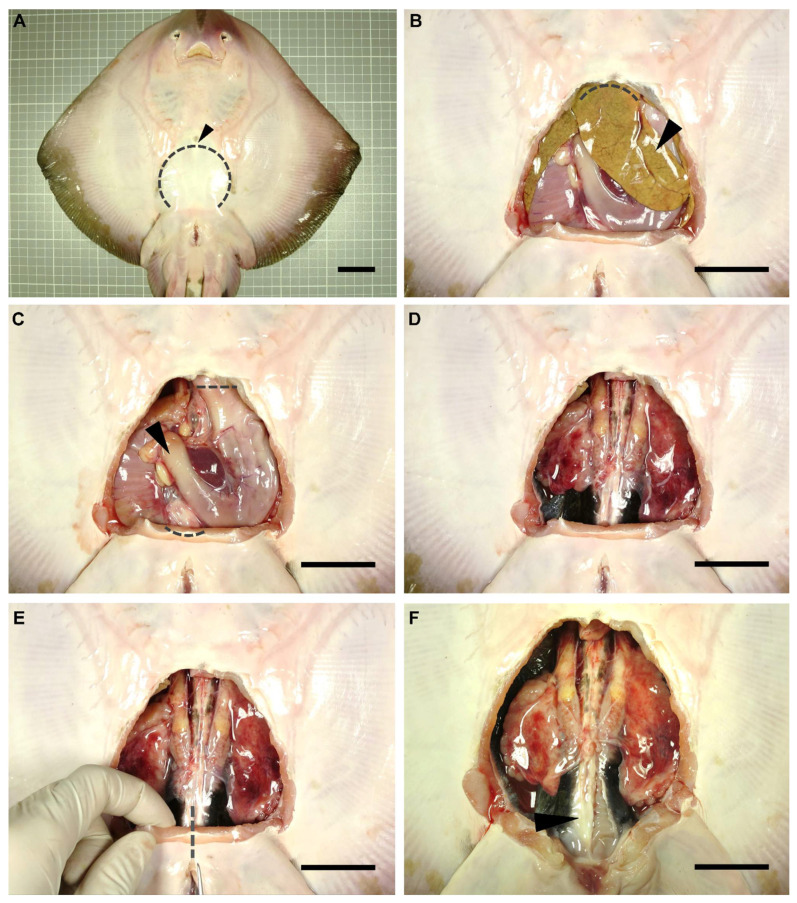
Dissection procedure. Relevant steps in the dissection procedure of a rough skate (*Raja radula*) male to reach the reproductive system (**A**). Removal of the abdominal wall (skin, muscle, parietal peritoneum) to gain access to the pleuroperitoneal cavity (**B**). Extraction of the liver (**C**) and digestive system (**D**). Incision of the puboischiadic bar (**E**) to fully expose the cloaca and the reproductive system (**F**). Arrowhead (**A**): coracoid bar; (**B**): liver; (**C**): digestive system; (**F**): parietal peritoneum. Dotted lines mark incision areas. Dotted line (**A**): abdominal wall; (**B**): cranial liver attachments; (**C**): oesophagus and rectum; (**E**): puboischiadic bar. Scale bar indicates 4 cm.

**Figure 2 animals-11-01888-f002:**
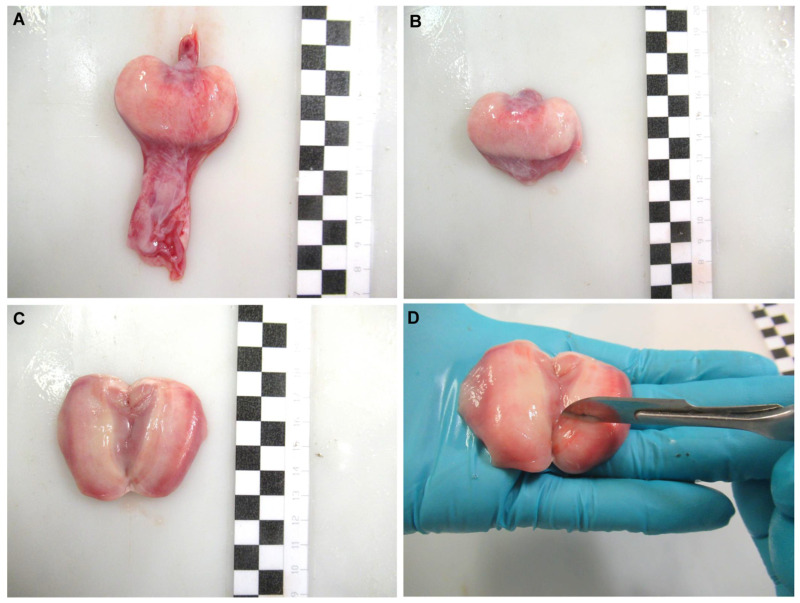
Sperm recovery from the oviducal gland. Oviducal gland from longnosed skate *Dipturus oxyrinchus* (**A**) removed from the oviduct. The gland was cleaned (**B)** and split through its longitudinal axis, exposing its lumen (**C**). A scraping was done over its luminal epithelium using the blunt edge of a scalpel (**D**) to collect a pearly mucus containing sperm. Each black and white square of the scale bar indicates 1 cm.

**Figure 3 animals-11-01888-f003:**
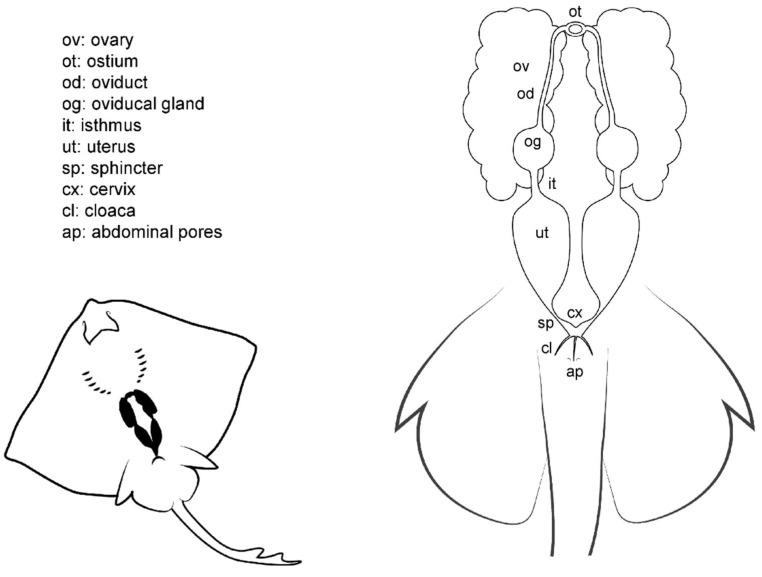
Female general anatomy. Morphological scheme of an ideal batoid female, showing the main reproductive structures: ovaries, oviducts, ostium, oviducal glands, isthmus, uteri, uterine sphincter, cervix, cloaca and abdominal pores.

**Figure 4 animals-11-01888-f004:**
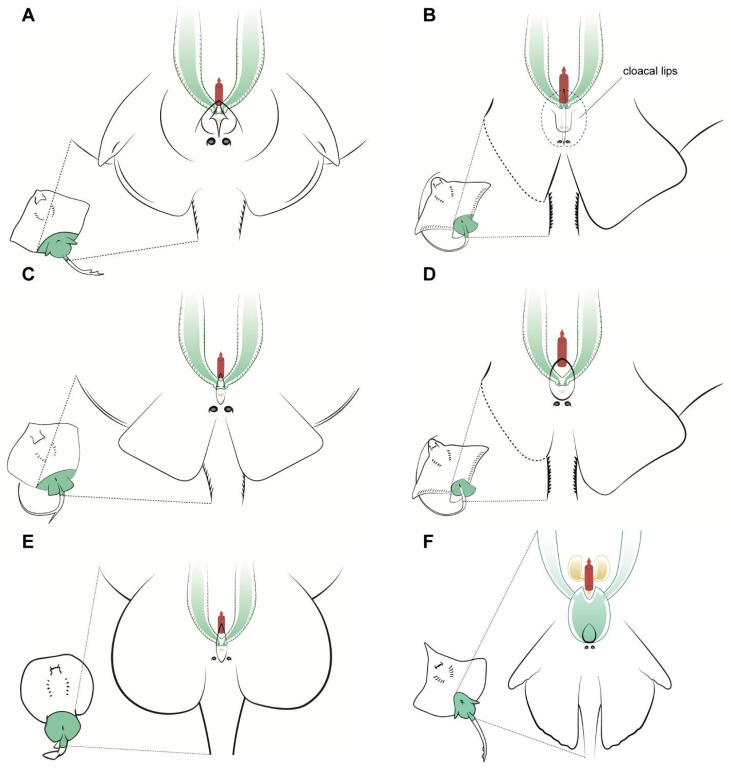
Specific female anatomy. Species-specific morphologies in female batoids that could be relevant when performing artificial insemination. Reproductive, excretory, and digestive systems are marked with different colours. Green: the uteri; yellow: the excretory system; red: access to the digestive system. The grey circles are the abdominal pores that communicate the pleuroperitoneal cavity with the exterior. (**A**) model for skates, genus *Raja*; (**B**) common eagle ray *Myliobatis Aquila*; (**C**) model observed in common stingray *Dasyatis pastinaca*; (**D**) bull ray *Aetomylaeus bovinus*; (**E**) marbled electric ray *Torpedo marmorata*; (**F**) longnosed skate *Dipturus oxyrinchus*.

**Figure 5 animals-11-01888-f005:**
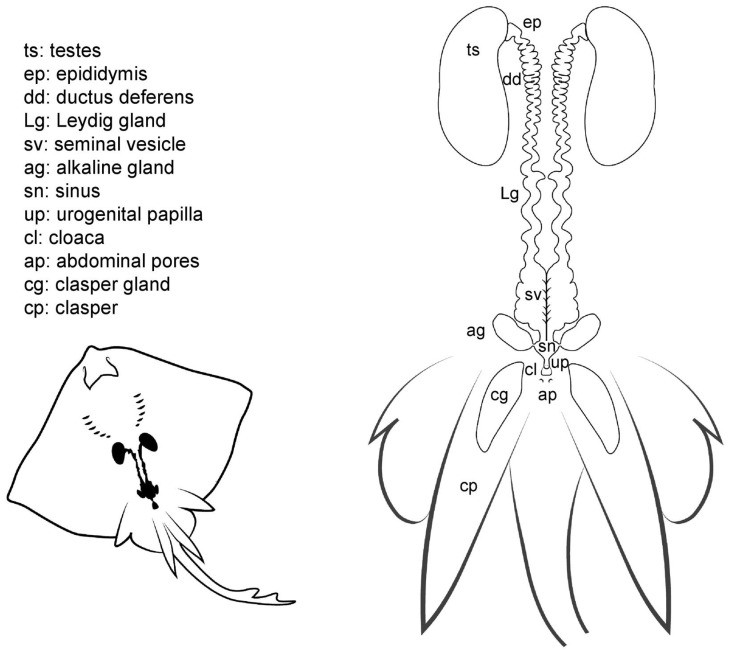
Male general anatomy. Morphological scheme of an ideal batoid male, showing the main reproductive structures: testes, epididymis, ductus deferens, Leydig seminal vesicles, alkaline glands, sinus, urogenital papilla, cloaca, abdominal pores and clasper glands.

**Figure 6 animals-11-01888-f006:**
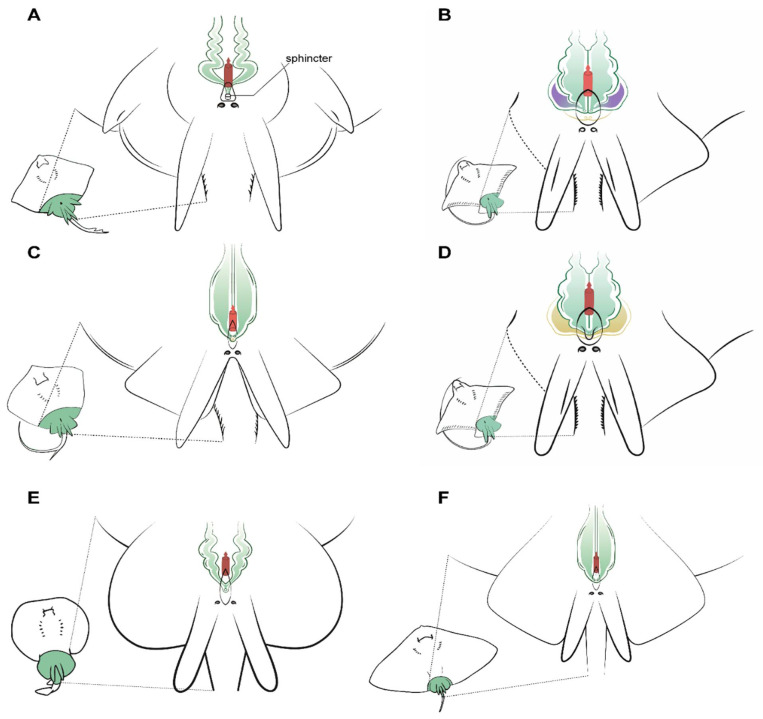
Specific male anatomy. Species-specific morphologies in batoids with relevance to performing sperm extraction, such as the pores leading to the seminal vesicles. Reproductive, excretory, and digestive systems are marked with different colours. Green: the seminal vesicle and ductus deferens; purple: the alkaline gland; yellow: the excretory system; red: the access to the digestive system. The grey circles are the abdominal pores that facilitate communication between the pleuroperitoneal cavity and the exterior. (**A**) model observed in genus *Raja*, with a unique sphincter on its urogenital papilla; (**B**) model observed in common stingray *Dasyatis pastinaca*, with two pores leading to the seminal vesicle and one pore to the urinary system; (**C**) model observed in common eagle ray *Myliobatis aquila* showing four different pores, with the two upper pores facilitating access to the seminal vesicle; (**D**) in bull ray *Aetomylaeus bovinus*, there are only two pores: one in the cranial part of the urogenital papilla leading to the seminal vesicle and one leading to the urinary system; (**E**) model observed in the marbled electric ray *Torpedo marmorata*, where there are two pores to access the seminal vesicles which are a widening of the ductus deferens; (**F**) the spiny butterfly ray *Gymnura altavela* reproductive system is similar to that of *D. pastinaca*, with two different pores to access the reproductive system.

**Table 1 animals-11-01888-t001:** Species in the study. Number of males (NM) and females (NF) from each species, size range of the specimens used and their origin and conservation status according to IUCN (International Union for Conservation of Nature) criteria for the Mediterranean [25]: least concern (LC), near threatened (NT), vulnerable (VU), endangered (EN) and critically endangered (CR). Animals from commercial fisheries were captured by gill net or bottom trawler and were available in fish markets (FM) or discarded as bycatch (BC). Animals from aquaria were part of the Oceanogràfic zoological collection (AQ).

Common Name	Scientific Name	NM	NF	IUCN	Source	Range (cm)
Rough skate	*Raja radula*	3	3	EN	FM	47–63
Spotted skate	*Raja montagui*	2	1	LC	FM	55–67
Mediterranean starry skate	*Raja asterias*	2	4	NT	AQ/FM	61–68
Thornback skate	*Raja clavata*	1	1	NT	FM	68–72
Undulate skate	*Raja undulata*	1		NT	AQ	86
Longnosed skate	*Dipturus oxyrinchus*	1	1	NT	BC	104–112
Spiny butterfly ray	*Gymnura altavela*	1		CR	BC	104
Common stingray	*Dasyatis pastinaca*	2	3	VU	BC	52–56
Common eagle ray	*Myliobatis aquila*	2	2	VU	BC	54–63
Bull ray	*Aetomylaeus bovinus*	2	2	CR	BC	89–103
Marbled electric ray	*Torpedo marmorata*	2	2	LC	BC	22–51

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
