# Peer review of "Reproductive Anatomy of Chondrichthyans: Notes on Specimen Handling and Sperm Extraction. I. Rays and Skates"

_animals, 2021, doi:10.3390/ani11071888_

Round 1

Reviewer 1 Report

This manuscript is a valuable contribution to the scientific literature.

My suggestions provided in comment boxes on the attached pdf file of the manuscript are mainly editorial. I appreciate that authors are not writing in their first language. 

Author Response

-Alter from "de" to 'the'.

It has been modified in the text, thanks for noticing.

 -Alter from "yolk sack" to 'yolk sac'.

It has been modified through the text.

-Alter from "which particular" to 'with particular' or 'which particularly".

It has been modified in the text as:

The main objective of this study is to provide a useful guide of the anatomy of the reproductive system of batoids, with particular focus on sperm….”

-I think the term "shelled" here will confuse readers.

It has been modified in the text:

“Animals from commercial fisheries were captured by gill net or bottom trawler and were available in fish markets.”

-I suggest altering from "necropsy" to 'dissection'. The term "necropsy" is normally used when dissecting an animal to determine the cause of death. Note that if the term is altered here, it should also be altered in the caption of Figure 1, paragraph 2 of section 2.3, and elsewhere.

Thanks for the suggestion, it has been modified through the text.

-The term "nidamental gland" is used in the early sections of the manuscript but alters to "oviducal gland" in the later sections. This might confuse some readers. Note that the more recent literature tends to apply the term "oviducal gland".

Indeed, we used the term nidamental gland in an earlier draft of the text, but we changed to “oviducal” following Hamlett 2005. It has been modified through the text.

-I suggest altering from "cloaca. While in" to 'cloaca, whereas in' or to ''cloaca. In'.

It has been modified in the text following your suggestion:

“In some species there are two separated orifices, one for each uterus, easily accessible from the cloaca, whereas in other species the orifices converge in a common cervix.”

-I suggest altering the term "eggshells" to 'egg cases'.

It has been modified through the text following your suggestion.

-I suggest altering from "is responsible" to 'functions'.

It has been changed following your suggestion:

“The functions of the gland are the storage of sperm, oocyte fertilization, and the formation of the egg case.”

-I feel the expression "The gland also supposes the terminological division of the oviduct in posterior or anterior oviduct" is confusing. Figure 3 shows the oviduct as being anterior to the oviducal gland and there is no "posterior oviduct". Consider 'In some species, the gland divides the oviduct into posterior and anterior oviducts.'

As you comment, the phrase can be confusing when compared to the figure. It has been changed following your suggestion:

In some species, the gland also supposes the terminological division of the oviduct in posterior or anterior oviduct.

-Consider altering from "two old groups [34]. Both groups have the same disposition of" to 'two plesiomorphic groups [34]. Both groups share'.

The sentence has been rewriting following your suggestion and the suggestion of another reviewer to:

“This anatomical resemblance was particularly remarkable between skates (species from the genus Raja) (Figure 4A) and the marbled electric ray Torpedo marmorata (Figure E), probably because of the close evolutionary relationship between these two plesiomorphic groups.”

-Consider altering "genus" to 'the genera'.

It has been changed following your suggestion.

-I suggest altering from "eggshell" to 'egg case' (also in last line on this page) and from "yolk-sack" to 'yolk-sac'.

Both terms have been changed through the text.

-I suggest altering from "two Myliobatiformes" to 'two species of Myliobatiformes'.

It has been changed following your suggestion.

-Alter "specie" to 'species'.

It has been modified in the text.

-The expression "the cervix are fused" is confusing. Perhaps the authors mean 'the paired cervices'; note that "cervices" also appear in the literature as 'cervixes'. An elasmobranch cervix is usually associated with each uterus. This ambiguity persists into the paragraph of section 3.3.

We changed the text following your suggestion. As you say, what we mean is that the cervices (of each uteri) are fused into a common cervix. We also changed the next section:

“In short, insemination is achieved by inserting a catheter through the cloaca to deliver sperm into the cervix or through the uterus, into the oviducal gland, although it has been proposed that intrauterine (or oviducal) insemination results in higher fertilization rates than cervix insemination.”

-The meaning of the expression "despite forming party" is unclear.

This was an error during writing. The phrase should be "are part", indicating that they belong to the same family. We have changed it to:

Animals from this genus possess considerably smaller eggs than those of the genus Dipturus, despite belonging to the same family: Rajidae [25].

-The term "sting ray" appears as 'stingray' elsewhere in the manuscript.

It has been changed to stingray, as in the other sections of the manuscript.

-I suspect that the term "impulsion" should be 'propulsion'.

Indeed, it should be propulsion. We have changed in the text.

-Note that "there is not sphincter" should be 'there is not a sphincter' or 'there is no sphincter'.

It has been changed to:

“…but in this species there is no sphincter, and a single…”

-I suggest altering from "as both seminal vesicles" to 'as the paired seminal vesicles'.

It has been changed following your suggestion.

-I suggest altering from "both seminal vesicles" to 'the seminal vesicles'.

It has been changed following your suggestion.

-Note that it is usual to spell the genus name out in full at the beginning of a paragraph or sentence.

Thanks for the comment, it has been changed in the text and we will take it into account for future writings.

-I suggest replacing the term "orifices" with 'apertures' for the context here.

It has been changed following your suggestion.

-I suspect that the expression "Although the storage" should be 'The storage'.

We have deleted the word “although” in the text.

-I suggest altering "nidamental" to 'oviducal'. It is best not to vary technical terms.

It has been changed following your suggestion.

-I suggest altering from "to rely" to 'relying'.

It has been changed following your suggestion.

-The expression "the responsible" should be 'responsible'.

It has been changed following your suggestion.

Reviewer 2 Report

Review of  “Reproductive anatomy of Chondrichthyans: notes on specimen handling and sperm extraction. I. Rays and skates.

The general aim of the study is to develop techniques for artificial fertilization of rays and skates for conservational purposes.

A major part of the article consists of gross anatomical descriptions of the reproductive organs of several species of rays and skates.  

There are probably masses of literature on the general anatomy of the reproductive organs of Batoidea. Much of it is probably in old articles and in several languages. Nevertheless, the findings in the present article must carefully compared with what is already known, which is far from the case now. What agrees with earlier descriptions, what does not? The discussion ought to focus on what is new.

The article describes how sperm were sampled from living or more or less freshly dead skates and rays, most of them discarded as bycatch or bought at fish markets. However, as a technical note this is of no use unless there is information on how well the artificial fertilizations went. There is no such information in the MS. I guess that the authors have this information (if they don’t, all is worthless) and intend to publish that in a separate paper, but I do not think that such a dividing up is justified.      

Other points

The whole idea of using artificial insemination is of course only useful in species that could be kept in captivity. How is this with the different species used?     

The English is often strange. e.g.:

  1. 2, last line. Shelled? Sold?

  1. 9 first paragraph end: “forming party of the same family” ought to be: forming part

I recommend that someone with a really good knowledge of English, preferably a native speaker, should go over the text

Author Response

A major part of the article consists of gross anatomical descriptions of the reproductive organs of several species of rays and skates.  

There are probably masses of literature on the general anatomy of the reproductive organs of Batoidea. Much of it is probably in old articles and in several languages. Nevertheless, the findings in the present article must carefully compared with what is already known, which is far from the case now. What agrees with earlier descriptions, what does not? The discussion ought to focus on what is new.

As you said, it is true that there are old articles which offer the basic anatomy of some of these animals. But the objective of this work is on detailing those anatomic structures that will have interest when practicing specific procedures. On the other hand, the old literature that exists is sometimes difficult to obtain and as you said, it is written in several languages (mainly in German). In addition, practical issues (such as the number and position of the urogenital pores) are often excluded from this type of studies.

But we should insist (and it is specified in the text), the objective of our study is very precise. The article is intended to be useful to aquarists, veterinarians or other researchers who wish to extract sperm in an efficient and accurate manner, without affecting the animals more than necessary.

The article describes how sperm were sampled from living or more or less freshly dead skates and rays, most of them discarded as bycatch or bought at fish markets. However, as a technical note this is of no use unless there is information on how well the artificial fertilizations went. There is no such information in the MS. I guess that the authors have this information (if they don’t, all is worthless) and intend to publish that in a separate paper, but I do not think that such a dividing up is justified.      

Until the moment of submitting this manuscript, no successful artificial insemination had never been done in this group of species in the world, including our group.

The possibility of extracting viable gametes, in this case sperm, allows the achievement of other objectives, beyond immediate artificial insemination. For example, our research group is working with cell cryopreservation protocols, which require sperm in optimal conditions. The success of cryopreservation is affected by sperm quality, so the usefulness of proper sperm extraction protocols cannot be neglected. Moreover, there are other type of studies where sperm extraction protocols should be considered: i) physiology of spermatozoa and seminal plasma; ii) pattern and dynamics of movement and sperm displacement; iii) taxonomic value of spermatozoa morphology; iv) impact of environment (temperature, pH, drugs) on sperm quality over time, etc. Therefore, while it is true that artificial insemination requires good sperm extraction procedures, other projects can benefit from correct manipulation in the extraction of gametes.

Other points

 The whole idea of using artificial insemination is of course only useful in species that could be kept in captivity. How is this with the different species used?     

Although it is true that the insemination of species can be mainly useful in animals kept in captivity, it is not the only use of this technique. Insemination of wild females may be completely feasible, especially in populations of sparsely dispersed species or those with a population bias (by number of males or their efficacy) in sexes. In fact, population control, including artificial insemination, can be a valid management tool in certain situations.

On the other hand, the diversity of species kept in zoological collections is not negligible. All the species with which this project has been developed are present in zoological collections of public aquaria or research centres, so knowing more about them is essential to increase the sustainability of these facilities.

The English is often strange. e.g.:

  1. 2, last line. Shelled? Sold?

  1. 9 first paragraph end: “forming party of the same family” ought to be: forming part

These errors and other grammatical inaccuracies have been corrected in the text.

I recommend that someone with a really good knowledge of English, preferably a native speaker, should go over the text

Reviewer 3 Report

OVERALL COMMENT:

The manuscript of Pablo García-Salinas, Victor Gallego and Juan F. Asturiano provides a useful guide of the anatomy of the reproductive system of Batoids, focusing on sperm procurement procedures and proposing preliminary indications on the female anatomy to be considered during artificial insemination. The work is interesting, really well written and structured. Moreover, it gives an accurate overview about the reproductive anatomy, providing new knowledge about procedure for the Elasmobranchs conservation thought artificial insemination. I think that the data and procedures showed in this paper could be very useful for the Researchers in this field in the future.

I found only minor points to address before publication, see specific comments below.

SPECIFIC COMMENTS:

Tab 1: The caption is a little confused; it misses the abbreviation for specimens belonging to Aquaria. I suggest a more orderly caption, which follow the course of the table.

MATERIALS AND METHODS 2.3: The first part of the sub-paragraph and the main objective of each necroscopy should be moved to discussion or introduction. This part results as an argumentation of the Authors that should not be placed in material and methods section.

MATERIALS AND METHODS 2.4.2: A bracket was missed in the proposition “(Over the location of the seminal……

MATERIALS AND METHODS 2.4.3: For a better readability, I suggest to the Authors to unifying the contents of 2.4.3 sub-paragraph with 2.4.1, since both the parts discuss the postmortem sperm extraction.

Fig. 3: There is an error in the image relative to the abdominal pores abbreviation. The cp abbreviation in the image should be replaced by ap (see the legend).

Fig. 6: There are some imperfections in D and F images. The left clasper in both images are incomplete and in D image there is also an imperfection in the general view of the species.

Best regards

The Reviewer

Author Response

Tab 1: The caption is a little confused; it misses the abbreviation for specimens belonging to Aquaria. I suggest a more orderly caption, which follow the course of the table.

Thank you very much for noticing this detail. It has included the abbreviation for the animals belonging to aquaria as follows:

“Animals in from aquaria were part of the Oceanogràfic zoological collection (AQ)”

MATERIALS AND METHODS 2.3: The first part of the sub-paragraph and the main objective of each necroscopy should be moved to discussion or introduction. This part results as an argumentation of the Authors that should not be placed in material and methods section.

Following your suggestion, the sub-paragraph has been moved to the “Results and discussion” section.

MATERIALS AND METHODS 2.4.2: A bracket was missed in the proposition “(Over the location of the seminal……

It has been included in the text, thanks for noticing.

MATERIALS AND METHODS 2.4.3: For a better readability, I suggest to the Authors to unifying the contents of 2.4.3 sub-paragraph with 2.4.1, since both the parts discuss the postmortem sperm extraction.

We have restructured the sections following your suggestion. We think it makes more sense this way, thank you.

Fig. 3: There is an error in the image relative to the abdominal pores abbreviation. The cp abbreviation in the image should be replaced by ap (see the legend).

Indeed, it is a mistake. Abdominal pores were called celomic pores (cp) in an early version of the figure. We have replaced the abbreviation in the figure.  

Fig. 6: There are some imperfections in D and F images. The left clasper in both images are incomplete and in D image there is also an imperfection in the general view of the species.

We have corrected the imperfection in image F, thanks for noticing the mistake.

The incomplete clasper in images D and F was a conscious decision (you can see the dotted lines) to make the image clearer, but we have included a new version of them.

Reviewer 4 Report

It is a significant contribution that fills a fundamental gap in existing evidence and thereby advances the science in the conservation of endangered species of Chondrichthyans. It is well designed as sufficiently rigorous to enable real forward motion in the field. I accept after English editing.

Author Response

Thanks a lot for your comments.

We have modified specific English mistakes identified by the rest of reviewers.

Reviewer 5 Report

The manuscript is an excellent piece of applicative anatomy of Chondrichthyans. I congratulate the authors for the excellent, interesting and useful work. A few very minor changes are suggested below.  

“… and the Elasmobranchs, commonly named sharks and rays”. Change to: and the Elasmobranchs, which include sharks and rays

“…. accepted under de formal name of Batoidea”. Change to “under the formal name…..”

To access the reproductive system, the liver (arrowhead in Figure 1B) and the digestive system (arrowhead in Figure 1C)  were removed.  The digestive system includes more organs located in different body regions (including mouth), so I would suggest to specify which organs were removed.

“Hepatic lobes were drawn forward and removed by cutting through their anterior attachments (ducts and mesentery) ….”  1) I would change “anterior” to “cranial” and “posterior” to “caudal” throughout the manuscript. 2) the only duct(s) of the liver is/are the hepatic duct(s). 3) Mesentery is the peritoneal fold connecting the intestine to the abdominal wall.

 “..to avoid damage to the mesorchium holding the testes.” The words “holding the testes” can be removed.

It is not clear what “fishery residues” are.

Please remove comma and add semicolon after “pincers”; remove full stop and add semicolon after “pipette” and then remove “And”.

“In skates (species from the genus Raja) (Figure 4A) and the marbled electric ray Torpedo marmorata (Figure E) in particular, shared a similar reproductive system, probably because of the close evolutionary relation-ship between these two old groups [34]”. Please check the grammar of this sentence.

“Both groups have the same disposition of the basic anatomical features”.  Maybe structures instead of “features”?

“Also, in this specie…” Change “specie” to “species”

“The catheter needs to pass through several anatomical barriers (such as cloacal morphology, uterine sphincters or cervix)….”. Please rephrase: morphology is not a barrier.

“Moreover, other factors, such the functionality of the uterus mentioned above should be considered”. Maybe add “as” after “such”?

“Externally, male batoids have paired elongated claspers (also called myxopterygium)”. Myxopterygium is singular.

“..two paired testes..” Please remove “two”

“The upper pore on the tip of the papilla is the access to the reproductive system (seminal vesicle and alkaline gland) and the lower pore leads ” What do upper and lower mean here? Maybe cranial and caudal?

  1. marmorata seminal vesicles converged…” Turn to present tense.

Abdominal massage in flat batoids (Order Rajiformes, Order Myliobatiformes, Order Torpediniformes). I would remove the repeated word “Order”

“It is important to note that within the batoids we can find some of the most en-dangered marine animals on the planet, such are sawfishes and similar species such wedgefishes and guitarfishes [49–51]”. Please check the grammar of this sentence.

Author Response

The manuscript is an excellent piece of applicative anatomy of Chondrichthyans. I congratulate the authors for the excellent, interesting and useful work. A few very minor changes are suggested below. 

“… and the Elasmobranchs, commonly named sharks and rays”. Change to: and the Elasmobranchs, which include sharks and rays

It has been modified in the text following your suggestion.

“…. accepted under de formal name of Batoidea”. Change to “under the formal name…..”

It has been modified in the text following your suggestion.

To access the reproductive system, the liver (arrowhead in Figure 1B) and the digestive system (arrowhead in Figure 1C)  were removed.  The digestive system includes more organs located in different body regions (including mouth), so I would suggest to specify which organs were removed.

It has been modified in the text following your suggestion:

“To access the reproductive system, the liver (arrowhead in Figure 1B) and the esophagus, stomach, spiral valve, and rectum (arrowhead in Figure 1C) were removed.

“Hepatic lobes were drawn forward and removed by cutting through their anterior attachments (ducts and mesentery) ….”  1) I would change “anterior” to “cranial” and “posterior” to “caudal” throughout the manuscript. 2) the only duct(s) of the liver is/are the hepatic duct(s). 3) Mesentery is the peritoneal fold connecting the intestine to the abdominal wall.

  • The words have been changed in the text following your suggestion.
  • It has been modified in the text: “Hepatic lobes were drawn forward and removed by cutting through their cranial attachments (hepatic duct and falciform ligament)”
  • The word mesentery has been removed from the sentence.

 “..to avoid damage to the mesorchium holding the testes.” The words “holding the testes” can be removed.

It has been modified in the text following your suggestion:

It is not clear what “fishery residues” are.

It has been modified in the text:

“Animals were cleaned using marine water to remove mucus and other biological remains such as blood and fishery residues (mud and remains of other organisms which may be found in animals obtained from fisheries)”.

Please remove comma and add semicolon after “pincers”; remove full stop and add semicolon after “pipette” and then remove “And”.

It has been modified in the text.

“In skates (species from the genus Raja) (Figure 4A) and the marbled electric ray Torpedo marmorata (Figure E) in particular, shared a similar reproductive system, probably because of the close evolutionary relation-ship between these two old groups [34]”. Please check the grammar of this sentence.

It has been modified in the text as follows:

“This anatomical resemblance was particularly remarkable between skates (species from the genus Raja) (Figure 4A) and the marbled electric ray Torpedo marmorata (Figure E), probably because of the close evolutionary relationship between these two plesiomorphic groups.”

“Both groups have the same disposition of the basic anatomical features”.  Maybe structures instead of “features”?

The word has been changed in the text following your suggestion

“Also, in this specie…” Change “specie” to “species”

The word has been changed in the text following your suggestion

“The catheter needs to pass through several anatomical barriers (such as cloacal morphology, uterine sphincters or cervix)….”. Please rephrase: morphology is not a barrier.

It has been modified in the text as follows:

“To be able to deposit a sperm sample in the correct oviduct, the catheter needs to be inserted through different structures (such as cloaca, uterine sphincters or cervix) that can hamper the overall process.”

“Moreover, other factors, such the functionality of the uterus mentioned above should be considered”. Maybe add “as” after “such”?

The word has been added in the text.

“Externally, male batoids have paired elongated claspers (also called myxopterygium)”. Myxopterygium is singular.

It has been modified in the text following your suggestion.

“..two paired testes..” Please remove “two”

It has been modified in the text following your suggestion.

“The upper pore on the tip of the papilla is the access to the reproductive system (seminal vesicle and alkaline gland) and the lower pore leads ” What do upper and lower mean here? Maybe cranial and caudal?

Instead of having two pores next to each other ( · · ), in this species the pores are arranged one above and the other below ( : ).

It has been modified in the text as follows:

“A. bovinus has only two pores arranged longitudinally on the tip of the urogenital papilla. The pore located above is the access to the reproductive system (seminal vesicle and alkaline gland) while the other pore leads to the excretory system.”

marmorata seminal vesicles converged…” Turn to present tense.

It has been modified in the text

Abdominal massage in flat batoids (Order Rajiformes, Order Myliobatiformes, Order Torpediniformes). I would remove the repeated word “Order”

It has been modified in the text following your suggestion.

“It is important to note that within the batoids we can find some of the most en-dangered marine animals on the planet, such are sawfishes and similar species such wedgfishes and guitarfishes [49–51]”. Please check the grammar of this sentence

It has been modified in the text as follows:

“It should be noted that among the batoids, we can find some of the most threatened marine animals on the planet: the sawfishes, guitarfishes and wedgefishes”

Round 2

Reviewer 2 Report

I do not find the replies of the authors very convincing.

The main content of the MS are gross anatomical descriptions of the reproductive organs of rays and skates. As for any scientific article, earlier literature (even it is in German and difficult to obtain) should be cited and it should be clear what information confirms or contradicts previous results and what is new. Should the new knowledge be very limited (not clear in this case) it is dubious if publication is motivated at all.

I was apparently wrong in my assumption that the authors had results on artificial fertilizations. This means that the point of merging studies drops, but the practical applicability hangs in the air. This is even more so for species that are difficult to keep in aquaria. The idea of artificially fertilizing wild females of sparsely dispersed species does not seem realistic, far better to leave them alone.